# Immunoinformatics Identification of the Conserved and Cross-Reactive T-Cell Epitopes of SARS-CoV-2 with Human Common Cold Coronaviruses, SARS-CoV, MERS-CoV and Live Attenuated Vaccines Presented by HLA Alleles of Indonesian Population

**DOI:** 10.3390/v14112328

**Published:** 2022-10-24

**Authors:** Marsia Gustiananda, Vivi Julietta, Angelika Hermawan, Gabriella Gita Febriana, Rio Hermantara, Lidya Kristiani, Elizabeth Sidhartha, Richard Sutejo, David Agustriawan, Sita Andarini, Arli Aditya Parikesit

**Affiliations:** 1Department of Biomedicine, School of Life Sciences, Indonesia International Institute for Life Sciences, Jl. Pulomas Barat Kav 88, Jakarta 13210, Indonesia; 2Department of Bioinformatics, School of Life Sciences, Indonesia International Institute for Life Sciences, Jl. Pulomas Barat Kav 88, Jakarta 13210, Indonesia; 3Department of Pulmonology and Respiratory Medicine, Faculty of Medicine University of Indonesia, Persahabatan Hospital, Jl. Persahabatan Raya 1, Jakarta 13230, Indonesia

**Keywords:** SARS-CoV-2, immunoinformatics, T-cell epitopes, human common cold coronaviruses, cross-reactive T-cells, universal coronavirus vaccines

## Abstract

Reports on T-cell cross-reactivity against SARS-CoV-2 epitopes in unexposed individuals have been linked with prior exposure to the human common cold coronaviruses (HCCCs). Several studies suggested that cross-reactive T-cells response to live attenuated vaccines (LAVs) such as BCG (Bacillus Calmette–Guérin), OPV (Oral Polio Vaccine), and MMR (measles, mumps, and rubella) can limit the development and severity of COVID-19. This study aims to identify potential cross-reactivity between SARS-CoV-2, HCCCs, and LAVs in the context of T-cell epitopes peptides presented by HLA (Human Leukocyte Antigen) alleles of the Indonesian population. SARS-CoV-2 derived T-cell epitopes were predicted using immunoinformatics tools and assessed for their conservancy, variability, and population coverage. Two fully conserved epitopes with 100% similarity and nine heterologous epitopes with identical T-cell receptor (TCR) contact residues were identified from the ORF1ab fragment of SARS-CoV-2 and all HCCCs. Cross-reactive epitopes from various proteins of SARS-CoV-2 and LAVs were also identified (15 epitopes from BCG, 7 epitopes from MMR, but none from OPV). A majority of the identified epitopes were observed to belong to ORF1ab, further suggesting the vital role of ORF1ab in the coronaviruses family and suggesting it as a candidate for a potential universal coronavirus vaccine that protects against severe disease by inducing cell mediated immunity.

## 1. Introduction

Severe acute respiratory syndrome coronavirus (SARS-CoV-2) is a new variant of coronavirus that caused an ongoing pandemic of coronavirus disease 2019 (COVID-19) [1]. As of May 2022, approximately 519 million cases and 6.3 million deaths had been reported globally [2]. A variety of studies and research have been undertaken into both preventive and curative measures against the SARS-CoV-2 virus. SARS-CoV-2 is an RNA virus from betacoronavirus of Coronaviridae family. Its genome is a positive single-stranded RNA with a length of 26 to 32 kb, which encodes several structural and nonstructural proteins. SARS-CoV-2 is characterized by having spike proteins on the surface of the enveloped virus [3]. As of 18 May 2022, WHO reported a total of two currently circulating variants of concern (VOCs), three previously circulating VOCs, eight previously circulating variants of interest (VOI), and one variant under monitoring (VUMs) of SARS-CoV-2 [4]. Different variants manifest different symptoms with different levels of severity and transmissibility. Typical symptoms include cough, fever, and sore throat. It may also develop into respiratory tract disease, which now has become the biggest contributing factor to the mortality rate [5].

Belonging to the same Coronaviridae family, the human common cold coronaviruses have been going around for decades. However, unlike COVID-19, it is well-known that the common cold coronaviruses do not normally cause serious signs and symptoms. Some common cold coronaviruses infecting humans are 229E and NL63 from the alphacoronavirus, HKU1, and OC43 from the betacoronavirus [6]. 

Cross-reactivity refers to unanticipated reactivity towards another antigen that differs from the expected targets, possibly due to similar structural regions [7]. Several reports of SARS-CoV-2 cross-reactive T-cells in naïve individuals have been linked to prior exposure to HCCCs [8,9]. T-cell memory and reactivity might play some role in attenuating the disease severity [6,10]. Several studies have also suggested that LAVs, such as OPV, BCG, and MMR vaccines may offer nonspecific protection by strengthening trained innate immunity against other infections, including SARS-CoV-2 [11,12]. The protection provided by LAVs was also suggested at the adaptive immune level where there is similarity of potential immunogenic epitopes sequence with SARS-CoV-2 [7,13,14,15,16,17].

In order for T-cells to elicit responses, the T-cell receptor (TCR) has to recognize peptides that are presented by HLA molecules [18]. However, it is a well-known fact that HLA molecules are population specific. Thus, different populations may present or possess different sets of alleles [19]. To date, most of the T-cell epitope studies have originated from European and North American populations where the predominant HLA allotypes differ from Indonesian [20,21]. As we mentioned in our previous publication, the Indonesian HLA alleles are frequently neglected and not well-studied in research and published papers [22]. Thus, by primarily using HLA allotypes prevalent in Indonesia, this study hopes to gain more understanding and could be the starting point in identifying SARS-CoV-2 immunogenic epitopes and supporting cell-mediated immunity studies relevant for Indonesian populations. Moreover, in the process, identifying conserved T-cell epitopes throughout the coronavirus family might help generate a universal vaccine construct [23].

## 2. Materials and Methods

### 2.1. Coronavirus Proteome Sequence Retrieval

The protein sequences of coronaviruses were primarily retrieved from NCBI (https://www.ncbi.nlm.nih.gov) on 14 september 2021. The reference numbers of each sequence are as follows: NC_002645.1 (229E), NC_006577.2 (HKU1), NC_005831.2 (NL63), NC_006213.1 (OC43), and NC_045512.2 (SARS-CoV-2) [24]. From each sequence, the encoded proteins present in all strains, namely ORF1ab, S, E, M, and N proteins, were selected for the subsequent analysis. The reference sequence of SARS-CoV-2 (NC_045512.2) was primarily used for T-cell epitope prediction.

For additional analysis performed in this study, ORF1ab sequences of SARS-CoV (NC_004718.3), MERS-CoV (NC_019843.3) were retrieved through NCBI [24]. ORF1ab sequences of SARS-CoV-2 VOCs Alpha (B.1.1.7), Beta (B.1.351), Delta (B.1.617.2), and Gamma (P.1) were obtained from the previous study conducted by Gustiananda et al. (2021) [25] and used for the analysis. The sequence for omicron (B. 1.1.529) variants was retrieved through NCBI on 13 February 2022 from Asian countries only. The included sequence was filtered through several criteria, such as a complete sequence of 7093–7096 amino acids in length with no ambiguous character and human as the host. The FASTA file of the sequences of SARS-CoV-2 VOCs used in this study can be found in Appendix A.

### 2.2. HLA Alleles Predominant in Indonesia Identification

The HLA alleles predominant in the Indonesian population were identified through the Allele Frequency Net Database (http://allelefrequencies.net/) [26] which was accessed on 16 september 2021. The run was done using ‘HLA Classical Allele Freq Search’ options, with the country parameters set to ‘Indonesia’. The HLA Class I A, HLA Class I B, and HLA Class II DRB1 alleles with frequencies of more than 5% were chosen for further analysis.

### 2.3. Cytotoxic T Lymphocyte (CTL) Epitopes Prediction

Epitope binding to HLA Class I was predicted using netCTLpan 1.1 tools (https://services.healthtech.dtu.dk/service.php?NetCTLpan-1.1) which was accessed on 30 January 2022 [27]. Several parameters, including peptide length (9-mer), weight on C terminal cleavage (0.225), weight on TAP transport efficiency (0.025), and threshold for epitope identification (1.0), were set accordingly. The strong binder epitopes, denoted by ‘E’ with percent rank <1%, were selected. Promiscuous epitopes that bind to different HLA alleles were selected for further analysis.

The selected 9-mer CTL epitopes were then checked for their immunogenicity using the immunogenicity tool provided by IEDB (http://tools.iedb.org/immunogenicity/) which was accessed on 21 April 2022 [28]. Since the TCR contact residue of CTL epitopes is located in positions 3–8, residue numbers 1, 2, and C-terminal were selected to be masked in the available option. Peptides with positive immunogenicity were selected for further analysis.

### 2.4. Helper T Lymphocyte (HTL) Epitopes Prediction

Prediction of peptide binding to HLA Class II DRB1 was conducted using netMHCII-pan 4.0 tools (https://services.healthtech.dtu.dk/service.php?NetMHCIIpan-4.0) which was accessed on 3 November 2021 [29]. Several parameters, including peptide length (15-mer), the threshold for strong binder (1%), and the threshold for weak binder (5%), were set accordingly. In addition, the strong binder epitopes, denoted by ‘SB,’ were selected for further analysis. Like the CTL epitopes, promiscuous epitopes that bind to different HLA alleles were selected for further analysis.

The selected 15-mer HTL epitopes were checked for their IFN-γ inducing ability through the IFNepitope server (http://crdd.osdd.net/raghava/ifnepitope/predict.php) which was accessed on 29 April 2022 [30]. The HTL must be able to induce IFN-γ to ensure induction of an appropriate immune response. The algorithm chosen for the prediction is the Support Vector Machine (SVM) based, while the model is left as default as ‘IFN-gamma vs. Non-IFN-gamma.’ Peptides with positive IFN-γ scores were selected for further analysis.

### 2.5. IEDB Epitopes Validation

Experimentally known epitopes from different studies are cataloged and available to be checked through the Immune Epitope Database (IEDB) (http://www.iedb.org) which was accessed on 17 May 2022 [31]. The short-listed epitopes, both CTL from step 2.3 and HTL from step 2.4, were checked against the database as validation measures. The epitope source was specified as ‘SARS-CoV-2’, ‘Human’ as the host, and antigen according to each generated epitope (ORF1ab, S, E, M, and N). Peptides reported and validated in IEDB were usually prioritized to be included in subsequent studies.

### 2.6. Conservancy and Cross-Reactivity Analysis of SARS-CoV-2 T-Cell Epitopes with HCCCs

In order to analyze the cross-reactivity of SARS-CoV-2 with other human CCCs, the conservancy analysis was done using the IEDB epitope conservancy analysis tool (http://tools.iedb.org/conservancy/) which was accessed on 9 March 2022 [32]. The respective epitopes from each identified protein were run against the four human CCCs. As a result, the epitopes that share 100% similarity across all strains were identified. Information on the 100% conserved epitopes, such as location in the gene, proteins, and domains in all five different viruses, was obtained through NCBI (https://www.ncbi.nlm.nih.gov) which was accessed on 14 March 2022.

Additional conservancy analysis was performed against the fully conserved epitopes. For more defined and complete results, the conserved epitopes were again run against SARS-CoV, MERS-CoV, previous, and currently circulating SARS-CoV-2 VOCs. Information regarding the epitopes in different viruses was also gathered and compared.

Epitopes that possess different residues in non-critical positions still have the possibility of binding to the same HLA alleles, provided that the residues fulfill the HLA binding motif. The epitopes that have identical TCR contact residue might still be recognized by the same T-cell clone. The TCR contact residues of CTL epitopes are located in residue number 3–8, while HTL epitopes are located in residue number 2, 3, 5, 7, and 8. Therefore, epitopes that possessed identical TCR contact residues across five different viruses during the conservancy run were identified. The sequence identity cutoff value in IEDB epitope conservancy analysis tool was first set to 67% to ensure optimal recognition and binding [33]. Following that, epitopes with identical TCR contact residue were sorted. The identified epitopes were then subjected to another NetCTLpan 1.1 analysis (https://services.healthtech.dtu.dk/service.php?NetCTLpan-1.1) for CTL epitopes and NetMHCIIpan 4.0 analysis (https://services.healthtech.dtu.dk/service.php?NetMHCIIpan-4.0) for HTL epitopes [27,29]. Both servers were accessed on 2 May 2022. The parameters for each prediction were identical to steps 2.3 and 2.4. This step was conducted to ensure that the residue difference in the sequence indeed did not affect the epitopes’ binding ability to the HLA alleles.

### 2.7. Cross-Reactivity Analysis of SARS-CoV-2 T-Cell Epitopes with OPV, BCG, and MMR 

The cross-reactivity analysis was done by using the Blastp (Protein BLAST) from the NCBI server (https://blast.ncbi.nlm.nih.gov/Blast.cgi?PAGE=Proteins) which was accessed on 23 April 2022 to identify similar amino acid sequences between SARS-CoV-2 T-cells epitopes with the epitope sequences from OPV, BCG, and MMR. The epitopes sequences were inputted into the “Query sequence” box. Meanwhile, the taxon ID from each live attenuated vaccine (OPV, BCG, MMR) was inputted into the “Organism” box. The taxon ID used in the epitope cross reactivity analysis are as follows: Mycobacterium tuberculosis variant bovis (1765), Human poliovirus 1 (12080), Human poliovirus 2 (12083), Human poliovirus 3 (12086), Measles virus strain Schwarz (132487), Rubella virus (11041), and Mumps orthorubulavirus (2560602). All of the options in the “exclude” section were selected. Some settings in the “Algorithm parameters” section were adjusted, such as the “Expect threshold” being 30000, the “Word size” being 2, and the “Matrix” being changed to PAM30; the “Compositional adjustments” was set to no adjustments, and the option in the “Filter” section was clicked.

### 2.8. Cross-Reactivity of Predicted Epitopes with Human Peptides

Blastp analysis (http://blast.ncbi.nlm.nih.gov/ Blast.cgi) was conducted to find SARS-CoV-2 epitopes that match with human peptides. The website was accessed on 2 May 2022. The method was adapted from previous studies [22,25]. The Blastp algorithm parameter was set as follows: expect threshold 30000, word size 2, matrix PAM30, gap cost set to existence = 9 and extension = 1; the compositional parameter was set to no adjustment, while the low complexity filter was disabled and automatically adjusted for short input sequences. The results from Blastp analysis were transferred into Microsoft Excel and were screened for the peptides which shared at least contiguously 7 identical amino acid residues with the human peptides with no gap and no mismatches residue.

### 2.9. Population Coverage Analysis

In order to assess whether the cross-reactive and short-listed epitopes are prevalent in Indonesia, population coverage analysis was performed through the IEDB population coverage analysis tools (http://tools.iedb.org/population/) which was accessed on 25 July 2022 [34]. The epitopes and corresponding HLA alleles that bind were inputted. Then, several parameters, including the population (Indonesia) and calculation options (Class I separate, Class II separate, and Class I and II combined), were set accordingly. The population coverage was then further analyzed.

### 2.10. Epitope Selection for Universal Coronavirus Vaccine Construct

Following different analyses, epitopes with attractive traits to be included in the coronavirus vaccine construct were further sorted. Criteria included conservancy, promiscuity, immunogenicity, IFNγ score, population coverage, no similarity with human peptides and IEDB reported peptides.

## 3. Results

### 3.1. HLA Allele Frequencies of Indonesian Population

Through the allele frequencies database, a total of 21 alleles from both HLA Class I and II with frequencies equal to or more than 5% were identified and used for the epitopes prediction (Figure 1). Out of 21, 15 of them belonged to HLA Class I alleles where six belonged to Class A and nine belonged to Class B. The highest frequencies from HLA Class I alleles were found to be HLA-A*24:07, with average frequencies of 23.1% coming from three different Indonesian populations. However, the highest frequency allele was located among the remaining six alleles that belonged to the HLA Class II DRB1 alleles. The HLA-DRB1*12:02 followed by HLA-DRB1*15:02 recorded an average frequency of 34.3% and 28.32%, coming from eight and six different Indonesian populations, respectively, making them the two highest frequency alleles of the HLA in the Indonesian population identified in this study.

### 3.2. Prediction of CTL Epitopes from SARS-CoV-2 Proteins

The prediction of CTL epitopes from SARS-CoV-2 protein with the HLA Class I alleles (%Rank <1%) resulted in 1599 bindings combined from 5 different SARS-CoV-2 proteins. In the process, it was found that HLA-B*15:21 and HLA-A*11:01 showed the most significant number of bindings, at 136 and 133 epitopes, respectively. Overall, as shown in Figure 2, each HLA Class I allele is predicted to bind to a large number of epitopes; the least records a total of 75 bindings, by HLA-B*44:03. Moreover, around 80% of the bindings from each allele were contributed by ORF1ab epitopes (Figure 2; blue color), followed by spike (S), membrane (M), nucleocapsid (N), and envelope (E) protein.

From the NetCTLpan 1.1 prediction, promiscuous peptides from SARS-CoV-2 (ORF1ab, S, E, M, and N) were further sorted and checked for their immunogenicity and population coverage. Promiscuous peptides here are defined as peptides that can bind with at least two different HLA Class I alleles. Out of 1599 predicted peptides, 414 of them are promiscuous. Highly promiscuous peptides that bind to at least five different HLA Class I alleles are listed in Table 1, while the complete list of peptides identified in this study can be found in Appendix A. Most of the peptides were observed to belong to the ORF1ab. With starting residue number 1767, the VMYMCTLSY peptide belonging to ORF1ab binds to 9 different peptides (HLA-A*11:01, HLA-A*34:01, HLA-B*15:02, HLA-B*15:13, HLA-B*15:21, HLA-B*18:01, HLA-B*35:05, HLA-B*44:03, HLA-B*58:01), making it the most promiscuous CTL peptide among other residues and covering the largest proportion in the Indonesian population of up to 90.22%.

Besides binding affinity to the HLA molecules, immunogenicity is an essential determinant in peptide recognition by TCR. Highly immunogenic CTL epitopes with a score >0.35 are listed in Table 2. A total of 14 highly immunogenic CTL epitopes (score >0.35) were identified along with their HLA Class I that binds, population coverage, and previously reported IEDB validation. The DYVYNPFMI epitope from ORF1ab recorded the highest immunogenicity score of 0.56221, bound to two HLA Class I alleles, and covered 58.77% of the Indonesian population. The binding with HLA-A*24:02 was also experimentally proven by HLA ligand assay and reported in IEDB.

### 3.3. Prediction of HTL Epitopes from SARS-CoV-2 Proteins

Despite the number of HLA Class II alleles being almost one-third the number of HLA Class I alleles, they record a nearly similar total number of predicted epitope bindings (Figure 3). The HLA-DRB1*15:02 allele holds the highest record, with a total of 299 HTL epitopes predicted to bind to it. In contrast, the HLA-DRB1*11:01 recorded a total of 170 bindings, the lowest record in HLA Class II alleles. All of the HLA Class II DRB1 included in this prediction binds to many more peptides when compared to HLA Class I. Moreover, consistent with the HLA Class I, ORF1ab is still responsible for most of the bindings (Figure 3; blue color), approximately 80% of the total. This is followed, in order, by spike, nucleocapsid, membrane, and envelope protein.

Following the NetMHCIIpan 4.0 prediction, promiscuous HTL peptides from SARS-CoV-2 (ORF1ab, S, E, M, and N) were also sorted and further checked for their IFN-γ score and population coverage. Similar to class I, promiscuous peptides here are defined as peptides that can bind to at least two different HLA Class II alleles. Out of 1422 predicted peptides, 414 of them are found to be promiscuous. The highly promiscuous peptides that bind to at least 3 out of 6 HLA Class II alleles are listed in Table 3, while the complete list of HTL peptides identified in this study can be found in Appendix A. With starting residue number 6751, the 15-mer peptides of LDDFVEIIKSQDLSV belonging to ORF1ab bind to all six different HLA Class II alleles (HLA-DRB1*07:01, HLA-DRB1*11:01, HLA-DRB1*12:02, HLA-DRB1*15:01, HLA-DRB1*15:02, HLA-DRB1*16:02), making it the most promiscuous HTL peptide that covers 95.26% of the Indonesian population. The binding between the peptides and HLA alleles has also been experimentally proven by T-cell assay and reported in IEDB.

Apart from binding affinity, IFN-γ is an important cytokine produced by HTL to trigger immune responses. Therefore, HTL epitopes with good IFN-γ inducing ability reflected as IFN-γ score are more likely to produce immune responses. Thus, 15 top 15-mer HTL epitopes with the highest IFN-γ score are listed in Table 4, along with information on HLA Class II that binds, population coverage, and previous IEDB reports. The complete list can also be found in Appendix A. The highest IFN-γ score was IAQFAPSASAFFGMS epitope from ORF9 that binds to DRB1_0701 with a score of 1.06167. The binding has also been experimentally proven by B-cell assay and reported in IEDB.

### 3.4. Identification of Conserved Epitopes in SARS-CoV-2 and Human CCCs

Cross-reactivity analysis was performed through the IEDB Conservancy Analysis Tools to identify conserved epitopes throughout the SARS-CoV-2 and four other human CCCs. The analysis was performed using five proteins (ORF1ab, S, E, M, and N) conserved in all five viruses against HLA Class I and II alleles. It was found that only two 9-mer epitopes were conserved in all five viruses (Table 5). Both epitopes belong to the ORF1ab, specifically within the region of nsp12, which encodes for the RNA dependent RNA polymerase (RdRp) enzyme of coronaviruses. The two epitopes bind to a total of 4 different HLA Class I alleles, with a population coverage of 13.07%. Moreover, specifically for the ‘SLAIDAYPL’ epitope, its binding with HLA-A*02:01 has been experimentally proven by both the T-cell assay and HLA ligand assay reported in IEDB.

Additional analysis was performed with the two identified conserved epitopes. The conserved epitopes were run against the SARS-CoV-2 current and previously circulating VOCs. The current VOCs include delta and omicron variants with a total of 1157 and 884 isolates, respectively. The previously circulating VOCs include alpha, beta, and gamma variants with a total of 158, 374, and 9 isolates, respectively. The sequences of SARS-CoV-2 variants can be found in Appendix A. From the conservancy run, it was found that all VOCs of SARS-CoV-2 possessed the two conserved epitope sequences (Table 6). These were slightly different in position but still belonging to the same SARS-CoV-like RdRp region and domain. The results show that the epitopes are fully conserved across SARS-CoV-2 variants despite different mutations that are responsible for emergence of different variants.

In addition to the VOCs, the epitopes were also run against the SARS-CoV and MERS-CoV from the same betacoronavirus family (Table 7). It was found that the SLAIDAYPL epitopes were conserved in both SARS-CoV and MERS-CoV. The HEFCSQHTM epitope was conserved 100% in SARS-CoV but not in MERS-CoV. One amino acid difference in the non-critical ninth residues was found in MERS-CoV, a change from methionine (M) to leucine (L). Despite the difference, the epitope from MERS-CoV still binds to the same HLA allele as SARS-CoV-2. Together with the SARS-CoV and MERS-CoV, the conservancy analysis indicated that the two identified epitopes are conserved across all coronaviruses that have infected humans.

### 3.5. Epitopes with Identical TCR Contact Residue between SARS-CoV-2 and Human CCC

Besides the fully conserved epitopes, epitopes with identical TCR contact residues between SARS-CoV-2 and four human CCCs were also identified. As previously indicated, epitopes with residue differences may still experience binding to the same HLA if the residues still fulfill the HLA binding motifs. The peptide which has an identical TCR contact residue might be recognized by the same T-cell clone. This study specifically used a sequence similarity cutoff value of 67% to ensure optimal recognition and binding [33]. Moreover, to ensure that the epitopes from human CCC with differing residues still indeed bind to the same HLA alleles with the original epitopes from SARS-CoV-2 reference sequence, another round of netCTLpan 1.1 analysis for 9-mer epitopes and netMHCIIpan 4.0 analysis for 15-mer epitopes was conducted.

During the process, a total of four 9-mer peptides (Table 8) and five 15-mer peptides (Table 9) from ORF1ab were identified. Almost all of them possess some residue difference in the non-TCR contact residue. Overall, however, more residue differences were observed in the 15-mer HTL epitopes. Data showed that not all listed peptides turn out to bind to the same alleles where the peptide from SARS-CoV-2 reference sequence binds, which could be due to the differences in the anchor residue of the peptide. The identified 9 peptides were found to mainly originate from nsp12, nsp13, and nsp14 of ORF1ab, which is suggested to play an essential role in the viral survival for it to retain some degree of conservancy across all five viruses. In addition, some of the bindings between the epitopes and HLA have also been reported in IEDB and validated through HLA ligand assay or B cell assay.

### 3.6. Cross-Reactive T-Cell Epitopes between SARS-CoV-2 and BCG

Potential cross-reactive CD8+ and CD4+ T-cells epitopes were discovered utilizing Blastp analysis and reinspected using NetCTLpan 1.1 and NetMHCIIpan-4.0 correspondingly. In total, the BCG vaccine contains 14 similar sequences as SARS-CoV-2 in ORF2, ORF3a, ORF4, ORF5, ORF6, ORF7b, ORF8, and ORF9 regions (Table 10). These similar sequences are not completely identical but have 2 to 4 amino acid substitutions. Despite that, these 14 epitopes are still able to bind to various HLA Class I alleles, especially HLA-A*02:01 and HLA-A*02:03. Moreover, several cross-reactive epitopes from BCG are also capable of binding different HLA alleles than the SARS-CoV-2 epitopes. Additionally, a few cross-reactive BCG epitopes also demonstrate stronger binding to HLA than SARS-CoV-2 epitopes according to the HLA binding affinity score.

Besides cross-reactive CTL epitopes, one cross-reactive HTL epitope was also identified through Blastp analysis. The cross-reactive HTL epitope from BCG is not identical to the SARS-CoV-2 epitope in the ORF7a region but has 2 amino acid substitutions in its core epitope. In spite of its difference, the cross-reactive HTL epitope is still able to bind HLA-DRB1*11:01 more strongly than SARS-CoV-2. In contrast, the cross-reactive HTL epitope has much lower Indonesian population coverage compared to the SARS-CoV-2 epitope (Table 11).

### 3.7. Cross-Reactive T-Cell Epitopes between SARS-CoV-2 and MMR

Similarly to BCG, several cross-reactive CTL epitopes were identified from the MMR vaccine, which is further divided into measles, mumps, and rubella. There were 7 similar sequences discovered from MMR compared with SARS-CoV-2 sequences in ORF2, ORF4, ORF7a, ORF7b, and ORF9 regions. These cross-reactive epitopes are able to bind to either one or two HLA Class I alleles, notably HLA-A*02:01 and HLA-A*02:03. However, only three cross-reactive epitopes from MMR demonstrate stronger HLA binding than SARS-CoV-2 epitopes, indicated by a lower HLA binding affinity score (Table 12).

### 3.8. Potential Epitope Set for Coronavirus Universal Vaccine Construct and Its Population Coverage

The conserved epitope from SARS-CoV-2 and HCCs identified in the preceding steps might be helpful in the universal vaccine construction. However, in regard to epitope selection for vaccines specifically, several additional criteria have to be considered, such as immunogenicity, IFN-γ inducing ability, population coverage, similarity with human peptides, and many more. Therefore, the selected epitopes were further characterized for the similarity with the human peptides and the entire epitope set was calculated for population coverage (Table 13).

Combining all the epitopes into a vaccine construct will result in 99.58% projected population coverage for Indonesia, with an average number of 4.25 epitope hits/HLA combinations recognized by the population, and a minimum number of 2.24 epitope hits/HLA combinations recognized by 90% of the population (Figure 4).

## 4. Discussion

T-cell responses to SARS-CoV-2 infection have been garnering attention lately due to potentially playing a vital role in measuring disease responses and progression, predicting disease recurrence, and control strategies [35]. Understanding T-cell responses can provide greater insights primarily on immune responses towards the disease, thus provide deeper understanding of its pathogenesis, vaccine development, responses, evaluation, disease severity, and much more [36]. There is strong evidence on T-cell cross-reactivity as the cause behind pre-existing immunity toward SARS-CoV-2 in the naïve population [8,9]. These cross-reactive immune responses reportedly correlate with shared homology between SARS-CoV-2 and the circulating HCoVs. Prior exposure to the human CCCs likely mediates the priming of SARS-CoV-2 responsive T-cells in the unexposed population [33,35,37,38].

This study discovered two fully conserved SARS-CoV-2 derived epitopes with 100% identity shared by all four common cold coronaviruses (HCoV-229E, HCoV-NL63, HCoV-OC43, HCoV-HKU1). They are presented by four major Indonesian HLA alleles (HLA-B*18:01, HLA-B*38:02, HLA-B*44:03 and HLA-A*02:01) with accumulated population coverage of 50.2%. The ‘SLAIDAYPL’ epitope binding with HLA-A*02:01, specifically, has been reported in IEDB. Moreover, the epitope has also reported to be conserved across CCCs in other population-specific HLA (UK and US) such as HLA-B*39 and HLA-B*62 [39].

In addition to the homologous epitopes, nine heterologous epitopes with identical TCR contact residue were also identified. They are all represented by at least one major Indonesian HLA allele and some of their bindings have been reported in IEDB. This result showed that cross-reactivity could indeed potentially arise due to sequence homology shared between the SARS-CoV-2 and human CCC. Previous studies demonstrated that T-cells, especially CD4+ T-cells that recognize human CCCs, exhibited cross-reactivity towards the homologous epitopes of SARS-CoV-2 [33]. It is important to note that the small number of epitopes identified in this study was most likely due to the analysis being performed across all four human CCC at once instead of towards each human CCC virus individually. More extensive epitope sequence similarity would be expected if the latter exercise was performed. A previous study identified as many as 13 fully conserved epitopes and 17 heterologous epitopes of SARS-CoV-2-derived epitopes when only compared to the betacoronavirus (HCoV-OC43 and HCoV-HKU1) [40]. Other studies have also consistently explored and reported a larger number of bindings when they conduct the analysis separately. Collectively, the data support the idea that cross-reactivity occurrence happens due to the shared homology of the HCoVs [39,41].

A further look into the fully conserved epitopes also showed that they are conserved in the previous and currently circulating SARS-CoV-2 VOCs, SARS-CoV and MERS-CoV. This result shows that there are low chances of these epitopes being responsible for the difference in disease outcomes and severity, given that their presence across the SARS-CoV-2 variants, SARS-CoV, MERS-CoV, and human CCCs is consistent. While data is still lacking to draw a conclusive claim, the information on the HLA alleles that bind to these conserved epitopes could possibly help in understanding different disease outcomes across the coronavirus family. The region where the conserved epitopes originated might play an essential role in the coronavirus family, for they are conserved across all the HCoVs.

Out of five proteins (ORF1ab, S, E, M, and N) used during the analysis, all of the identified shared epitopes in this study belong to the ORF1ab. This may indicate that ORF1ab possessed a high degree of conservancy in the coronavirus family. This finding is also aligned with our previous study which validated it through entropy analysis [25] and studies done by others using genome-wide screening technology [42]. Besides the shared epitopes, ORF1ab also ranks highest in the number of predicted epitopes when compared to the rest of the proteins such as S, E, M, and N. This is understandable given the number of epitopes highly correlated with the gene length, as ORF1ab length is almost two-thirds of the whole genome [40]. This data was corroborated by others which revealed that ORF1ab contains the highest number of experimentally proven immunodominant epitopes [42] and highlighted the role of ORF1ab as the region with the most cross-reactive T-cells that can elicit responses in individuals exposed to other HCoVs but naive to SARS-CoV-2 infection [38].

Moreover, ORF1ab also has a primary function as a replicase enzyme that is vital for viral survival [43]. The ORF1ab is further cleaved into 16 nsps, where several of them have an integral function during the viral life cycle [44]. This study specifically identified epitopes belonging to the nsp12, nsp13, and nsp14 regions. The nsp12 encodes for RdRp protein responsible for viral replication, transcription, and overall survival. The nsp13 belongs to the helicase superfamily, which possesses both helicase and NTPase activity [45]. They primarily function to unwind the DNA or RNA strands during replication; they also support and interact with other replication-transcription complexes [46]. Lastly, the nsp14 encodes for proteins necessary for proofreading ability, evading the host’s antiviral responses, and ensuring mRNA stability and translation [47]. The essential functions of each region in SARS-CoV-2 further support the idea of them being conserved across the whole coronavirus family. Previous studies have reported that nsp12-nsp14 are among the most conserved and accessible sites in the entire SARS-CoV-2 proteome, which is aligned with this study’s results [45,46]. As a conserved region, they also become an attractive target for intervention such as drug targets or vaccines [47].

The role of live attenuated vaccines such as BCG, OPV and MMR in attenuating COVID-19 diseases has been hypothesized [48] and investigated in some clinical trials [49,50]. In addition, a case-control study was also performed which investigated the MMR vaccination’s probable protective effect against SARS-CoV-2 [51]. Although most clinical and case-control studies focus on trained innate immunity, some studies showed that these LAVs also contain cross-reactive T-cells epitopes similar to epitopes of SARS-CoV-2 that may generate adaptive immunological memory [7,16,17]. These cross-reactive T-cells epitopes might be one of the main factors contributing to protective effects from LAVs apart from the trained innate immune system. The Blastp analysis conducted in this study discovered that BCG and MMR contain 15 and 7 similar yet not identical sequences with SARS-CoV-2, respectively. Despite the fact that these sequences have 2 to 4 amino acid substitution, these sequences are potentially able to induce cross-reactive CTL against SARS-CoV-2. Additionally, one sequence discovered from BCG also has the ability to give rise to cross-reactive HLT against SARS-CoV-2. However, the dissimilarity in terms of which HLAs bind to the cross-reactive epitopes suggests that not all people will benefit from the presence of cross-reactive epitopes. Hence, only people who have certain HLA alleles will be able to respond to the epitopes and gain benefit from the heterologous immunity.

In contrast to other studies [13,14] the Blastp analysis did not find any cross-reactive CTL and HTL epitopes from OPV specific to the Indonesian population. One of the factors that might lead to no cross-reactive epitopes between OPV and SARS-CoV-2 is the genome nature of poliovirus. Compared to other viruses, the genomic RNA of the poliovirus is only approximately 7500 nucleotides in length. Hence, the poliovirus is classified as a small virus in comparison to BCG and MMR, resulting in lower possibilities of similarity matches in Blastp analysis. However, the result of this study does not imply that OPV does not have beneficial cross-protection effects on the Indonesian population. There is still the possibility that OPV provides nonspecific protection through trained innate immunity, such as Toll Like Receptor (TLR) stimulation, notably the TLR3/TLR7, against SARS-CoV-2 which eventually leads to the early activation of innate immunity, such as macrophages and dendritic cells [52] as well as the potential cross-reactivity of the humoral immune responses [15].

Regarding the association of the T-cell cross-reactivity with the clinical outcomes, there are limitations on the conclusions that can be drawn from immunoinformatics analyses. The ideas discussed here require investigation and validation through laboratory experiments. Several studies, for instance, proved that cross-reactive memory T-cells indeed result in protection by measuring the IL-2 and IFN-γ secretion through fluorescence-linked immunospot (FLISpot) assay on PBMC [40]. An experiment using an epitope-mapping and flow cytometry activation-induced marker assay suggested that the presence of cross-reactive T-cells might be responsible for the different clinical outcomes and severity in COVID-19 patients [33].

It is still difficult and too early for now to conclude whether this cross-reactivity indeed results in cross-protection or instead results in enhanced pathogenicity. Apart from T-cell reactivity, the association of HLA alleles with COVID-19 severity has also been reported [53]. HLA genetic background is considered one of the contributing factors to different SARS-CoV-2 infection outcomes (transmissibility, susceptibility, progression, and severity), as it is directly related to an individual’s immunogenetic variation [21]. Each allele possesses a different level of binding affinity. This binding affinity, as mentioned previously, will directly affect and produce different immune responses such as neutralizing antibodies or pro-inflammatory cytokines. In general, most of the HLA Class II alleles are less significant when compared to HLA Class I in regard to severity [54].

Out of 21 HLA alleles prevalent in Indonesia used in this study (Figure 1), only some have been further analyzed for their contribution towards severity in different populations (not Indonesian). For example, HLA-A*11:01 was proven by different studies to be highly correlated with severe disease and outcomes in the European and Chinese populations [21,54]. One study implied that individuals who possessed HLA-A*11:01 and HLA-A*24:02 might produce better T-cell immune responses when compared to HLA*02:01 [55]. HLA-A*02:01 instead is associated with an increased risk of infection due to its lower ability to present SARS-CoV-2 antigen in multinational populations. Lastly, the DRB1*12:01 are found to display protective effects against SARS-CoV-2, while the DRB1*15:01 are found to be increased in asymptomatic patients in the European and Chinese population [20,54]. Findings on different populations may or may not be reflected in the Indonesian population. However, it can be seen that differences in the genetic profile and geographic background will affect the outcomes of infection along with other factors such as age, gender, ABO blood groups, and other comorbid conditions [20,41].

The COVID-19 pandemic has once again shown the ability of viruses from the coronavirus family to cause massive outbreaks [56,57]. Studies have highlighted SARS-CoV-2 potential to evolve, survive, and reemerge, owing to its efficient host-switching and genetic recombination ability [58,59]. The immediate example that can be witnessed is the emergence of numerous SARS-CoV-2 variants with different transmissibility and immune evasion ability, which influences the treatment efficiency, including for vaccines. These factors stress the need for a universal coronavirus vaccine in order to minimize disease burden or even prevent the next coronavirus outbreak [56].

This study suggested some epitopes that might be helpful in the universal coronavirus vaccine construction (Table 13). Based on a series of analyses conducted in this study, ORF1ab was specifically identified as an attractive candidate for a universal vaccine. Epitopes belonging to the ORF1ab family have some features of high conservancy across the coronavirus family, promiscuousness that covers a large population, immunodominance, and good IFN-γ inducing ability; these make them very promising as possible targets for universal coronavirus vaccines. Several studies that primarily utilized ORF1ab for their multi-epitope vaccines design have also professed interest in and highlighted ORF1ab qualities as a vaccine target [25,60]. The presence of pre-existing cross-reactive memory T-cells recognizing epitopes from the early proteins that were made by the infected cells such as the polymerase proteins have been associated with the abortive infection which is beneficial in protecting individual from SARS-CoV-2 infection [61]. This supports ORF1ab as a candidate for a universal coronavirus vaccine and shows the benefit of cross-reactive T-cells. Yet there are still reports of contradictory findings where it is believed exposure to human common cold coronavirus does not primarily explain the cross-reactivity [62]. Debates over the exact cause and mechanism are still circulating. Pre-existing immunity may result in either protection or increased susceptibility toward SARS-CoV-2 [24]. Regardless of different opinions and arguments, understanding T-cell responses and reactivity is believed to be the key to understanding the disease responses to ensure suitable and effective control strategies are implemented [35,36].

To date, no studies have conducted extensive research on the universal coronavirus vaccine. However, some essential points can be learned from the attempt to make a universal influenza vaccine [63]. It is vital to ensure that a universal vaccine offers broad and durable protection that is effective towards at least three-fourths of viruses from the family, despite the mutations that may arise over time [64,65]. It also must be suitable for all age groups and possess good immunogenicity, low allergenicity, and low homology with the human peptides [64]. A better understanding of the infection itself will also help improve and speed up the vaccine design process, assessing the vaccine responses and evaluation [64]. All in all, it may take a long time to actually produce an effective and safe universal vaccine, but scientists believe the idea of a universal coronavirus vaccine is possible with further exploration through research and analysis [56,57,65].

## 5. Conclusions

Through immunoinformatics approaches, two fully conserved SARS-CoV-2 derived epitopes with 100% sequence similarity shared by all four human CCCs, along with additional nine heterologous epitopes with identical TCR contact residues and cross-reactive epitopes with the live attenuated vaccines were identified. Each epitope was found to be represented by at least one major Indonesian HLA allele, covering a considerable population with good binding affinity. These findings overall support the idea that T-cell cross reactivity occurrence in SARS-CoV-2 naïve individuals could arise from shared sequence homology between SARS-CoV-2 and human CCCs. Additional analysis performed with SARS-CoV, MERS-CoV, and SARS-CoV-2 VOCs also indicated that the conserved epitopes are not responsible for differential outcomes in regards to the resulting signs, symptoms, and severity, as their presence is consistent across all assessed isolates. Moreover, with most predicted conserved epitopes belonging to the ORF1ab, this study also implies the vital role of ORF1ab in the coronavirus family. This further suggests its potential as a candidate for the universal coronavirus vaccine target. 

The present study is limited to in-silico analysis which cannot confirm matters concerning T-cell reactivity and HLA association with clinical outcomes and severity. Therefore, further validation using laboratory experiments that allow measurement of immune system key player (i.e., IL-2, IFN-γ) levels might provide greater insights to resolve the issues raised.

## Figures and Tables

**Figure 1 viruses-14-02328-f001:**
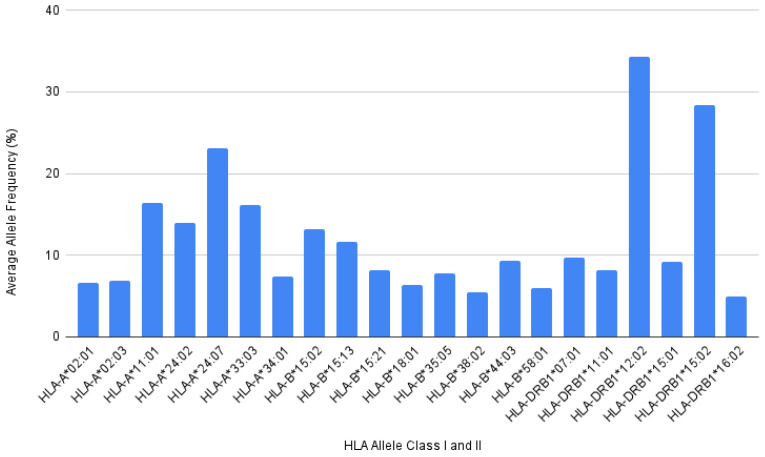
HLA Allele of Indonesian population with frequency equal or more than 5% identified from Allele Frequency Net Database. A total of 21 alleles were identified: 6 HLA Class I A, 8 HLA Class I B, and 6 HLA Class II DRB1 alleles.

**Figure 2 viruses-14-02328-f002:**
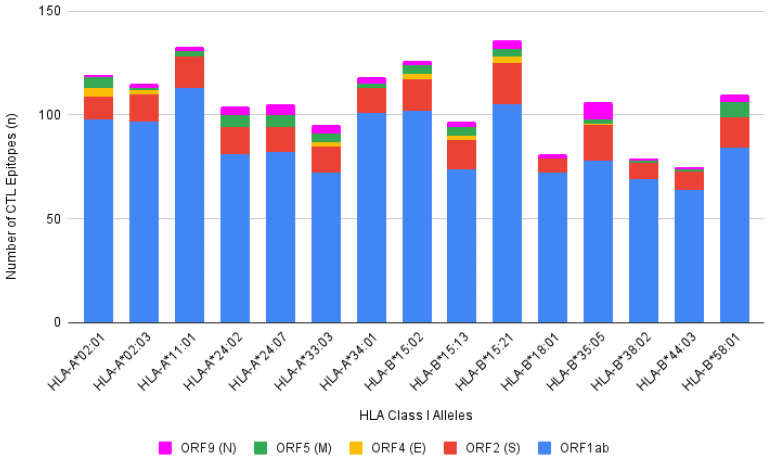
Number of predicted CTL epitopes with %Rank less than 1% from SARS-CoV-2 proteins (ORF1ab, ORF2 (S), ORF4 (E), ORF5 (M), and ORF9 (N)) with previously identified HLA Class I alleles. A total of 1599 bindings were predicted through NetCTLpan 1.1 with the highest binding recorded by HLA-B*15:21 and the least by HLA-B*44:03, 136 and 75, respectively. ORF1ab is represented by blue color; ORF2 (S) is represented by red color; ORF4 (E) is represented by yellow color; ORF5 (M) is represented by green color; ORF9 (N) is represented by pink color.

**Figure 3 viruses-14-02328-f003:**
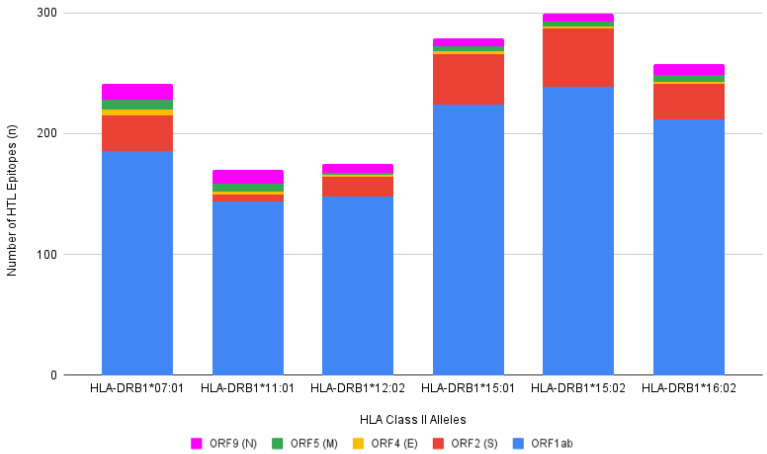
Number of predicted HTL epitopes binding with %Rank less than 1% from SARS-CoV-2 proteins (ORF1ab, ORF2 (S), ORF4 (E), ORF5 (M), and ORF9 (N)) with previously identified HLA Class II DRB1 alleles. A total of 1422 bindings were predicted through NetMHCpan 4.0 with the highest binding recorded by HLA-DRB1*15:02 and the least by HLA-DRB1*11:01, 299 and 170, respectively. ORF1ab is represented by blue color; ORF2 (S) is represented by red color; ORF4 (E) is represented by yellow color; ORF5 (M) is represented by green color; ORF9 (N) is represented by pink color.

**Figure 4 viruses-14-02328-f004:**
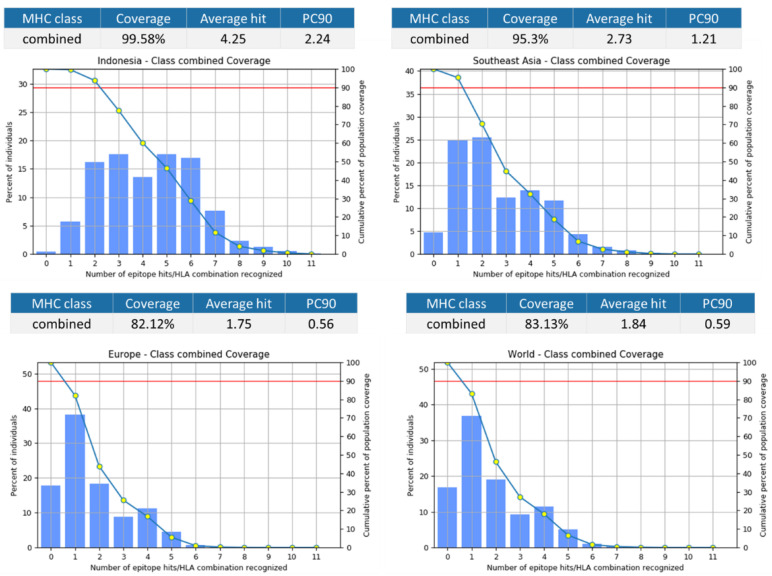
Projected coverage of the vaccine construct containing all epitopes in Table 10 for Indonesian, South East Asia, Europe, and the world population.

**Table 1 viruses-14-02328-t001:** Highly promiscuous 9-mer CTL epitopes from five different SARS-CoV-2 proteins (ORF1ab, S, E, M, and N) that bind to at least five different HLA Class I alleles. Immunogenicity, population coverage, and previous report of IEDB finding of each peptide are also specified.

Protein	Start Residue	Peptide	HLA Class I Allele	Immunogenicity	Population Coverage (%)
ORF1ab	899	WSMATYYLF	HLA-A*24:02^c^,HLA-A*24:07, HLA-B*15:02, HLA-B*15:13, HLA-B*15:21, HLA-B*18:01, HLA-B*35:05, HLA-B*58:01	0.00742	89.83
ORF1ab	915	LASHMYCSF	HLA-B*15:02, HLA-B*15:13, HLA-B*15:21, HLA-B*35:05, HLA-B*58:01	−0.12588	68.68
ORF1ab	1269	LVSDIDITF	HLA-B*15:02, HLA-B*15:13, HLA-B*15:21, HLA-B*35:05, HLA-B*58:01	0.2281	68.68
ORF1ab	1581	QVVDMSMTY	HLA-A*34:01, HLA-B*15:02, HLA-B*15:13, HLA-B*15:21, HLA-B*35:05	−0.1306	67.15
ORF1ab	1585	MSMTYGQQF	HLA-B*15:02, HLA-B*15:13, HLA-B*15:21, HLA-B*35:05, HLA-B*58:01	−0.24791	68.68
ORF1ab	1767	VMYMGTLSY	HLA-A*11:01, HLA-A*34:01, HLA-B*15:02, HLA-B*15:13, HLA-B*15:21, HLA-B*18:01, HLA-B*35:05, HLA-B*44:03, HLA-B*58:01	−0.03831	90.22
ORF1ab	1805	MMSAPPAQY	HLA-B*15:02, HLA-B*15:13, HLA-B*15:21, HLA-B*35:05, HLA-B*58:01,	0.08593	68.68
ORF1ab	1822	CASEYTGNY	HLA-A*34:01, HLA-B*15:02, HLA-B*15:13, HLA-B*15:21, HLA-B*35:05	0.18319	67.15
ORF1ab	2253	MSNLGMPSY	HLA-A*11:01, HLA-A*34:01, HLA-B*15:02, HLA-B*15:13, HLA-B*15:21, HLA-B*35:05, HLA-B*58:01	0.05143	81.53
ORF1ab	2272	STNVTIATY	HLA-A*11:01, HLA-A*34:01, HLA-B*15:02, HLA-B*15:13, HLA-B*15:21, HLA-B*58:01	−0.00384	75.56
ORF1ab	2322	LVAEWFLAY	HLA-A*11:01, HLA-A*34:01, HLA-B*15:02, HLA-B*15:13, HLA-B*15:21, HLA-B*18:01, HLA-B*35:05	−0.27126	81.9
ORF1ab	2381	RMYIFFASF	HLA-A*24:02, HLA-A*24:07, HLA-B*15:02, HLA-B*15:13, HLA-B*15:21, HLA-B*35:05, HLA-B*38:02, HLA-B*58:01	0.08761	89.49
ORF1ab	2383	YIFFASFYY	HLA-A*11:01, HLA-A*34:01, HLA-B*15:02, HLA-B*15:13, HLA-B*15:21, HLA-B*18:01, HLA-B*35:05	0.3343	81.9
ORF1ab	2593	YVNTFSSTF	HLA-A*34:01, HLA-B*15:02, HLA-B*15:13, HLA-B*15:21, HLA-B*35:05	0.22019	67.15
ORF1ab	2936	NVLEGSVAY	HLA-A*34:01, HLA-B*15:02, HLA-B*15:21, HLA-B*18:01, HLA-B*35:05	−0.1059	61.73
ORF1ab	3135	VPFWITIAY	HLA-B*15:02, HLA-B*15:13, HLA-B*15:21, HLA-B*18:01, HLA-B*35:05	−0.05933	69.31
ORF1ab	3372	QTFSVLACY	HLA-A*11:01, HLA-A*34:01, HLA-B*15:02, HLA-B*15:13, HLA-B*15:21, HLA-B*58:01	−0.06765	75.56
ORF1ab	3645	TVAYFNMVY	HLA-A*11:01, HLA-A*34:01, HLA-B*15:02, HLA-B*15:21, HLA-B*35:05	−0.01723	68.69
ORF1ab	3710	LMNVLTLVY	HLA-B*15:02, HLA-B*15:13, HLA-B*15:21, HLA-B*35:05, HLA-B*58:01	−0.01717	68.68
ORF1ab	3945	SEFSSLPSY ^b^	HLA-B*15:02, HLA-B*15:13, HLA-B*15:21, HLA-B*18:01^b^, HLA-B*44:03^c^	0.12602	69.92
ORF1ab	3948	SSLPSYAAF	HLA-A*24:02, HLA-A*24:07, HLA-B*15:02, HLA-B*15:13, HLA-B*15:21, HLA-B*35:05, HLA-B*58:01	−0.22896	87.09
ORF1ab	4263	STVLSFCAF	HLA-B*15:02, HLA-B*15:13, HLA-B*15:21, HLA-B*35:05, HLA-B*58:01	0.15992	68.68
ORF1ab	4271	FAVDAAKAY	HLA-A*34:01, HLA-B*15:02, HLA-B*15:13, HLA-B*15:21, HLA-B*18:01^b^, HLA-B*35:05	0.03414	73.5
ORF1ab	4757	LSFKELLVY	HLA-B*15:02, HLA-B*15:13, HLA-B*15:21, HLA-B*35:05, HLA-B*58:01	0.08254	68.68
ORF1ab	4904	RLYYDSMSY ^a^	HLA-A*11:01, HLA-B*15:02, HLA-B*15:13, HLA-B*15:21, HLA-B*35:05	0.06159	73.23
ORF1ab	5057	MVMCGGSLY	HLA-A*11:01, HLA-A*34:01, HLA-B*15:02, HLA-B*15:13, HLA-B*15:21, HLA-B*35:05	−0.12346	77.56
ORF1ab	5245	LMIERFVSL	HLA-A*02:01^c^, HLA-A*02:03, HLA-B*15:02, HLA-B*15:21, HLA-B*38:02	−0.11314	54.17
ORF1ab	5267	EYADVFHLY	HLA-A*24:02, HLA-A*24:07, HLA-A*33:03, HLA-A*34:01, HLA-B*18:01	0.00446	85.01
ORF1ab	5532	VVYRGTTTY ^a^	HLA-A*11:01, HLA-A*34:01, HLA-B*15:02, HLA-B*15:13, HLA-B*15:21, HLA-B*35:05, HLA-B*58:01	−0.21535	81.53
ORF1ab	5678	YVFCTVNAL ^a^	HLA-A*02:01, HLA-A*02:03, HLA-A*34:01, HLA-B*15:02, HLA-B*15:21, HLA-B*35:05, HLA-B*38:02	0.24593	69.29
ORF1ab	5980	SMMGFKMNY	HLA-A*11:01, HLA-A*34:01, HLA-B*15:02, HLA-B*15:13, HLA-B*15:21, HLA-B*35:05	−0.11462	77.56
ORF1ab	6153	HSIGFDYVY	HLA-A*34:01, HLA-B*15:02, HLA-B*15:13, HLA-B*15:21, HLA-B*18:01, HLA-B*35:05, HLA-B*58:01	−0.21438	78.69
ORF1ab	6424	MMISAGFSL ^a^	HLA-A*02:01^c^, HLA-A*02:03, HLA-B*15:02, HLA-B*15:13, HLA-B*15:21, HLA-B*35:05, HLA-B*38:02	0.38891	74.8
ORF1ab	6433	WVYKQFDTY	HLA-A*34:01, HLA-B*15:02, HLA-B*15:13, HLA-B*15:21, HLA-B*35:05	0.06486	67.15
ORF1ab	7000	EAFLIGCNY	HLA-A*34:01, HLA-B*15:02, HLA-B*15:13, HLA-B*15:21, HLA-B*35:05	0.07781	67.15
ORF1ab	7019	YVMHANYIF	HLA-A*24:02, HLA-A*24:07, HLA-B*15:02, HLA-B*15:13, HLA-B*15:21, HLA-B*35:05	−0.12171	89.49
ORF2 (S)	83	LPFNDGVYF	HLA-B*15:02, HLA-B*15:13, HLA-B*15:21, HLA-B*18:01, HLA-B*35:05	0.11767	69.31
ORF2 (S)	159	YSSANNCTF	HLA-A*24:07, HLA-B*15:02, HLA-B*15:13, HLA-B*15:21, HLA-B*35:05, HLA-B*58:01	−0.04954	80.83
ORF2 (S)	161	SANNCTFEY	HLA-A*11:01, HLA-B*15:02, HLA-B*15:13, HLA-B*15:21, HLA-B*35:05, HLA-B*58:01	0.13273	77.97
ORF2 (S)	268	YLQPRTFLL ^a^	HLA-A*02:01^c^, HLA-A*02:03, HLA-A*24:02^a^, HLA-A*24:07, HLA-B*38:02	0.1305	74.4
ORF2 (S)	368	YNSASFSTF	HLA-A*24:02, HLA-A*24:07, HLA-B*15:02, HLA-B*15:13, HLA-B*15:21, HLA-B*35:05	−0.18217	89.49
ORF2 (S)	686	VASQSIIAY	HLA-B*15:02, HLA-B*15:13, HLA-B*15:21, HLA-B*35:05, HLA-B*58:01	−0.0709	68.68
ORF2 (S)	690	SIIAYTMSL ^a^	HLA-A*02:01^c^, HLA-A*02:03, HLA-A*34:01, HLA-B*15:21, HLA-B*38:02	−0.12935	48.2
ORF2 (S)	868	MIAQYTSAL ^a^	HLA-A*02:03^b^, HLA-A*34:01, HLA-B*15:02, HLA-B*15:21, HLA-B*35:05	−0.18768	57.95
ORF2 (S)	895	IPFAMQMAY	HLA-B*15:02, HLA-B*15:13, HLA-B*15:21, HLA-B*18:01, HLA-B*35:05	−0.32801	69.31
ORF5	169	VATSRTLSY ^b^	HLA-B*15:02, HLA-B*15:13, HLA-B*15:21, HLA-B*35:05, HLA-B*58:01	−0.17295	68.68
ORF9	265	KAYNVTQAF ^b^	HLA-A*24:02, HLA-A*24:07, HLA-B*15:02, HLA-B*15:13, HLA-B*15:21, HLA-B*35:05, HLA-B*58:01	−0.00587	87.09

^a^ The binding between the peptide and HLA has been experimentally proven by T-cell assay and reported in IEDB; ^b^ The binding between the peptide and HLA has been experimentally proven by HLA ligand assay and reported in IEDB; ^c^ The binding between the peptide and HLA has been experimentally proven by both T-cell assay and HLA ligand assay and reported in IEDB.

**Table 2 viruses-14-02328-t002:** Highly immunogenic (>0.35) 9-mer CTL epitopes identified from five different SARS-CoV-2 proteins (ORF1ab, ORF2 (S), ORF5 (M), and ORF9 (N)).

Protein	Start Residue	Peptide	HLA Class I Allele	Immunogenicity	Population Coverage (%)
ORF1ab	44	HLKDGTCGL	HLA-A*02:03	0.40234	7.68
ORF1ab	1634	YHTTDPSFL	HLA-B*38:02	0.44289	10.65
ORF1ab	2001	TYKPNTWCI	HLA-A*24:02 ^b^; HLA-A*24:07	0.5572	58.77
ORF1ab	2789	ITPVHVMSK	HLA-A*11:01 ^b^	0.43221	29.64
ORF1ab	3541	RTILGSALL	HLA-B*58:01	0.45285	11.11
ORF1ab	3640	LPSLATVAY	HLA-B*15:02, HLA-B*15:21, HLA-B*18:01, HLA-B*35:05	0.35609	55.67
ORF1ab	4580	TVQFCDAMR	HLA-A*33:03	0.37218	29.13
ORF1ab	6158	DYVYNPFMI	HLA-A*24:02 ^b^, HLA-A*24:07	0.56221	58.77
ORF1ab	6819	LEKCDLQNY	HLA-B*18:01, HLA-B*44:03	0.37101	28.15
ORF2 (S)	1098	GTHWFVTQR	HLA-A*11:01 ^c^, HLA-A*33:03 ^b^	0.35133	53.67
ORF2 (S)	1208	YIKWPWYIW	HLA-B*58:01	0.42524	11.11
ORF2 (S)	1211	WPWYIWLGF	HLA-B*18:01, HLA-B*35:05	0.41673	27.22
ORF5 (M)	25	FLFLTWICL	HLA-A*02:01 ^a^	0.35397	13.07
ORF9 (N)	103	LSPRWYFYY	HLA-A*24:07, HLA-B*58:01	0.35734	45.59

^a^ The binding between the peptide and HLA has been experimentally proven by T-cell assay and reported in IEDB; ^b^ The binding between the peptide and HLA has been experimentally proven by HLA ligand assay and reported in IEDB; ^c^ The binding between the peptide and HLA has been experimentally proven by both T-cell assay and HLA ligand assay and reported in IEDB.

**Table 3 viruses-14-02328-t003:** Highly promiscuous 15-mer HTL Epitopes from five different SARS-CoV-2 proteins (ORF1ab, S, E, M, and N) that bind to at least three different HLA Class II alleles. IFN-γ score, population coverage, and previous report of IEDB finding of each peptide are also specified.

Protein	Start Residue	Peptide	HLA Class II Allele	IFN-γ Score	Population Coverage (%)
ORF1ab	457	EKVNINIVGDFKLNE	HLA-DRB1*12:02, HLA-DRB1*15:01, HLA-DRB1*15:02, HLA-DRB1*16:02	0.073961117	87.72
ORF1ab	458	KVNINIVGDFKLNEE ^c^	HLA-DRB1*12:02, HLA-DRB1*15:01, HLA-DRB1*15:02, HLA-DRB1*16:02	−0.10696324	87.72
ORF1ab	734	KAPKEIIFLEGETLP	HLA-DRB1*12:02, HLA-DRB1*15:01, HLA-DRB1*15:02, HLA-DRB1*16:02	−0.24929914	87.72
ORF1ab	735	APKEIIFLEGETLPT	HLA-DRB1*12:02, HLA-DRB1*15:01, HLA-DRB1*15:02, HLA-DRB1*16:02	0.082149378	87.72
ORF1ab	736	PKEIIFLEGETLPTE ^c^	HLA-DRB1*12:02, HLA-DRB1*15:01, HLA-DRB1*15:02, HLA-DRB1*16:02	0.077069314	87.72
ORF1ab	1053	KPTVVVNAANVYLKH	HLA-DRB1*12:02, HLA-DRB1*15:01, HLA-DRB1*15:02, HLA-DRB1*16:02	−0.45376782	87.72
ORF1ab	1171	TNVYLAVFDKNLYDK	HLA-DRB1*07:01, HLA-DRB1*12:02, HLA-DRB1*15:02, HLA-DRB1*16:02	−0.32342333	91.86
ORF1ab	1349	CKSAFYILPSIISNE	HLA-DRB1*11:01, HLA-DRB1*12:02, HLA-DRB1*15:02, HLA-DRB1*16:02	0.28976547	86.38
ORF1ab	1350	KSAFYILPSIISNEK ^a,c^	HLA-DRB1*11:01, HLA-DRB1*12:02, HLA-DRB1*15:02, HLA-DRB1*16:02	0.33784834	86.38
ORF1ab	1363	EKQEILGTVSWNLRE	HLA-DRB1*07:01, HLA-DRB1*15:01, HLA-DRB1*15:02, HLA-DRB1*16:02	−0.0096068911	65.21
ORF1ab	1419	GARFYFYTSKTTVAS	HLA-DRB1*07:01, HLA-DRB1*15:01, HLA-DRB1*15:02, HLA-DRB1*16:02	−0.48814565	65.21
ORF1ab	1799	QQESPFVMMSAPPAQ	HLA-DRB1*11:01, HLA-DRB1*12:02, HLA-DRB1*15:02, HLA-DRB1*16:02	−0.31609633	86.38
ORF1ab	1800	QESPFVMMSAPPAQY ^c^	HLA-DRB1*11:01, HLA-DRB1*12:02, HLA-DRB1*15:02, HLA-DRB1*16:02	−0.15478331	86.38
ORF1ab	1801	ESPFVMMSAPPAQYE ^a^	HLA-DRB1*11:01, HLA-DRB1*12:02, HLA-DRB1*15:02, HLA-DRB1*16:02	−0.21040286	86.38
ORF1ab	1802	SPFVMMSAPPAQYEL^c^	HLA-DRB1*11:01, HLA-DRB1*12:02, HLA-DRB1*15:02, HLA-DRB1*16:02	−0.18691649	86.38
ORF1ab	1908	TEQPIDLVPNQPYPN ^c^	HLA-DRB1*12:02, HLA-DRB1*15:01, HLA-DRB1*15:02, HLA-DRB1*16:02	−0.18762311	87.72
ORF1ab	1909	EQPIDLVPNQPYPNA	HLA-DRB1*12:02, HLA-DRB1*15:01, HLA-DRB1*15:02, HLA-DRB1*16:02	−0.15613773	87.72
ORF1ab	2211	LEASFNYLKSPNFSK ^a^	HLA-DRB1*07:01, HLA-DRB1*12:02, HLA-DRB1*15:02, HLA-DRB1*16:02	−0.12722255	91.86
ORF1ab	2212	EASFNYLKSPNFSKL ^c^	HLA-DRB1*07:01, HLA-DRB1*12:02, HLA-DRB1*15:02, HLA-DRB1*16:02	−0.37773341	91.86
ORF1ab	2518	LSHFVNLDNLRANNT ^c^	HLA-DRB1*11:01, HLA-DRB1*12:02, HLA-DRB1*15:02, HLA-DRB1*16:02	−0.29350215	86.38
ORF1ab	2958	DGSIIQFPNTYLEGS	HLA-DRB1*12:02, HLA-DRB1*15:01, HLA-DRB1*15:02, HLA-DRB1*16:02	0.087720744	87.72
ORF1ab	3150	TKHFYWFFSNYLKRR ^c^	HLA-DRB1*07:01, HLA-DRB1*15:01, HLA-DRB1*15:02, HLA-DRB1*16:02	−0.34583424	65.21
ORF1ab	3816	STQEFRYMNSQGLLP	HLA-DRB1*07:01, HLA-DRB1*12:02, HLA-DRB1*15:01, HLA-DRB1*15:02, HLA-DRB1*16:02	−0.28198599	94.14
ORF1ab	3817	TQEFRYMNSQGLLPP	HLA-DRB1*07:01, HLA-DRB1*12:02, HLA-DRB1*15:01, HLA-DRB1*15:02, HLA-DRB1*16:02	−0.41760878	94.14
ORF1ab	3818	QEFRYMNSQGLLPPK	HLA-DRB1*07:01, HLA-DRB1*12:02, HLA-DRB1*15:01, HLA-DRB1*15:02, HLA-DRB1*16:02	−0.25437902	94.14
ORF1ab	4559	ENPDILRVYANLGER	HLA-DRB1*12:02, HLA-DRB1*15:01, HLA-DRB1*15:02, HLA-DRB1*16:02	0.011725622	87.72
ORF1ab	4560	NPDILRVYANLGERV	HLA-DRB1*12:02, HLA-DRB1*15:01, HLA-DRB1*15:02, HLA-DRB1*16:02	0.22993566	87.72
ORF1ab	4561	PDILRVYANLGERVR ^a^	HLA-DRB1*12:02, HLA-DRB1*15:01, HLA-DRB1*15:02, HLA-DRB1*16:02	0.26161469	87.72
ORF1ab	5018	MPNMLRIMASLVLAR ^c^	HLA-DRB1*12:02, HLA-DRB1*15:01, HLA-DRB1*15:02, HLA-DRB1*16:02	0.32014953	87.72
ORF1ab	5019	PNMLRIMASLVLARK ^a^	HLA-DRB1*12:02, HLA-DRB1*15:01, HLA-DRB1*15:02, HLA-DRB1*16:02	0.39136883	87.72
ORF1ab	5166	GLVASIKNFKSVLYY ^a^	HLA-DRB1*12:02, HLA-DRB1*15:01, HLA-DRB1*15:02, HLA-DRB1*16:02	0.43106149	87.72
ORF1ab	5167	LVASIKNFKSVLYYQ	HLA-DRB1*12:02, HLA-DRB1*15:01, HLA-DRB1*15:02, HLA-DRB1*16:02	0.093707023	87.72
ORF1ab	5168	VASIKNFKSVLYYQN	HLA-DRB1*12:02, HLA-DRB1*15:01, HLA-DRB1*15:02, HLA-DRB1*16:02	0.29825337	87.72
ORF1ab	5169	ASIKNFKSVLYYQNN	HLA-DRB1*12:02, HLA-DRB1*15:01, HLA-DRB1*15:02, HLA-DRB1*16:02	0.25775961	87.72
ORF1ab	5450	CTERLKLFAAETLKA	HLA-DRB1*12:02, HLA-DRB1*15:01, HLA-DRB1*15:02, HLA-DRB1*16:02	−0.10920914	87.72
ORF1ab	5451	TERLKLFAAETLKAT	HLA-DRB1*07:01, HLA-DRB1*12:02, HLA-DRB1*15:01, HLA-DRB1*15:02, HLA-DRB1*16:02	0.013340883	94.14
ORF1ab	5769	PAEIVDTVSALVYDN ^c^	HLA-DRB1*07:01, HLA-DRB1*15:01, HLA-DRB1*15:02, HLA-DRB1*16:02	−0.31287444	65.21
ORF1ab	5845	VASKILGLPTQTVDS	HLA-DRB1*12:02, HLA-DRB1*15:01, HLA-DRB1*15:02, HLA-DRB1*16:02	0.015678357	87.72
ORF1ab	5846	ASKILGLPTQTVDSS	HLA-DRB1*12:02, HLA-DRB1*15:01, HLA-DRB1*15:02, HLA-DRB1*16:02	−0.12146879	87.72
ORF1ab	6002	EEAIRHVRAWIGFDV ^c^	HLA-DRB1*12:02, HLA-DRB1*15:01, HLA-DRB1*15:02, HLA-DRB1*16:02	0.42312165	87.72
ORF1ab	6039	GVNLVAVPTGYVDTP	HLA-DRB1*07:01, HLA-DRB1*12:02, HLA-DRB1*15:01, HLA-DRB1*15:02, HLA-DRB1*16:02	−0.46784679	94.14
ORF1ab	6087	VRIKIVQMLSDTLKN	HLA-DRB1*12:02, HLA-DRB1*15:01, HLA-DRB1*15:02, HLA-DRB1*16:02	−0.08846654	87.72
ORF1ab	6516	KPVPEVKILNNLGVD ^c^	HLA-DRB1*12:02, HLA-DRB1*15:01, HLA-DRB1*15:02, HLA-DRB1*16:02	−0.092818209	87.72
ORF1ab	6517	PVPEVKILNNLGVDI	HLA-DRB1*12:02, HLA-DRB1*15:01, HLA-DRB1*15:02, HLA-DRB1*16:02	−0.0068847756	87.72
ORF1ab	6518	VPEVKILNNLGVDIA	HLA-DRB1*12:02, HLA-DRB1*15:01, HLA-DRB1*15:02, HLA-DRB1*16:02	−0.25937664	87.72
ORF1ab	6519	PEVKILNNLGVDIAA	HLA-DRB1*12:02, HLA-DRB1*15:01, HLA-DRB1*15:02, HLA-DRB1*16:02	−0.41267881	87.72
ORF1ab	6750	LLDDFVEIIKSQDLS ^c^	HLA-DRB1*07:01, HLA-DRB1*11:01, HLA-DRB1*15:01, HLA-DRB1*15:02, HLA-DRB1*16:02	−0.27733231	68.03
ORF1ab	6751	LDDFVEIIKSQDLSV ^a^	HLA-DRB1*07:01, HLA-DRB1*11:01, HLA-DRB1*12:02, HLA-DRB1*15:01, HLA-DRB1*15:02, HLA-DRB1*16:02	−0.04841137	95.26
ORF1ab	6752	DDFVEIIKSQDLSVV ^c^	HLA-DRB1*07:01, HLA-DRB1*12:02, HLA-DRB1*15:01, HLA-DRB1*15:02, HLA-DRB1*16:02	0.28882695	94.14
ORF2 (S)	307	TVEKGIYQTSNFRVQ ^c^	HLA-DRB1*07:01, HLA-DRB1*15:01, HLA-DRB1*15:02, HLA-DRB1*16:02	0.45017873	65.21
ORF2 (S)	308	VEKGIYQTSNFRVQP ^b^	HLA-DRB1*07:01, HLA-DRB1*15:01, HLA-DRB1*15:02, HLA-DRB1*16:02	0.20827026	65.21
ORF2 (S)	309	EKGIYQTSNFRVQPT ^b,c^	HLA-DRB1*07:01, HLA-DRB1*15:01, HLA-DRB1*15:02, HLA-DRB1*16:02	0.46467579	65.21
ORF2 (S)	310	KGIYQTSNFRVQPTE ^b^	HLA-DRB1*07:01, HLA-DRB1*15:01, HLA-DRB1*15:02, HLA-DRB1*16:02	0.36164402	65.21
ORF4 (E)	58	VYSRVKNLNSSRVPD	HLA-DRB1*12:02, HLA-DRB1*15:01, HLA-DRB1*15:02, HLA-DRB1*16:02	−0.18705221	87.72
ORF4 (E)	59	YSRVKNLNSSRVPDL ^c^	HLA-DRB1*12:02, HLA-DRB1*15:01, HLA-DRB1*15:02, HLA-DRB1*16:02	−0.45329667	87.72
ORF5 (M)	163	DLPKEITVATSRTLS	HLA-DRB1*07:01, HLA-DRB1*15:01, HLA-DRB1*15:02, HLA-DRB1*16:02	−0.36193098	65.21
ORF5 (M)	164	LPKEITVATSRTLSY	HLA-DRB1*07:01, HLA-DRB1*12:02, HLA-DRB1*15:01, HLA-DRB1*15:02, HLA-DRB1*16:02	−0.17020992	94.14
ORF5 (M)	165	PKEITVATSRTLSYY ^a^	HLA-DRB1*07:01, HLA-DRB1*12:02, HLA-DRB1*15:01, HLA-DRB1*15:02, HLA-DRB1*16:02	0.2121047	94.14

^a^ The binding between the peptide and HLA has been experimentally proven by T-cell assay and reported in IEDB; ^b^ The binding between the peptide and HLA has been experimentally proven by HLA ligand assay and reported in IEDB; ^c^ The binding between the peptide and HLA has been experimentally proven by B-cell assay and reported in IEDB.

**Table 4 viruses-14-02328-t004:** Top 15 of 15-mer HTL epitopes with highest IFN-γ score from different SARS-CoV-2 proteins with %Rank less than 1%.

Protein	Start Residue	Peptide	HLA Class II Allele	IFN-γ Score	Population Coverage (%)
ORF9 (N)	304	IAQFAPSASAFFGMS ^b^	HLA-DRB1*07:01	1.0616777	20.49
ORF9 (N)	309	PSASAFFGMSRIGME	HLA-DRB1*11:01	0.94891009	4.83
ORF9 (N)	303	QIAQFAPSASAFFGM	HLA-DRB1*07:01	0.83655244	20.49
ORF1ab	7076	GRLIIRENNRVVISS	HLA-DRB1*15:01, HLA-DRB1*15:02, HLA-DRB1*16:02	0.79853065	51.25
ORF9 (N)	310	SASAFFGMSRIGMEV	HLA-DRB1*11:01	0.79707565	4.83
ORF1ab	7082	ENNRVVISSDVLVNN ^b^	HLA-DRB1*07:01, HLA-DRB1*15:01, HLA-DRB1*15:02	0.79231845	61.9
ORF1ab	7075	KGRLIIRENNRVVIS	HLA-DRB1*15:01, HLA-DRB1*15:02, HLA-DRB1*16:02	0.78951146	51.25
ORF1ab	4124	WPLIVTALRANSAVK	HLA-DRB1*11:01, HLA-DRB1*1202	0.78569358	60.59
ORF1ab	1631	AFEYYHTTDPSFLGR	HLA-DRB1*07:01	0.77956564	20.49
ORF1ab	1630	EAFEYYHTTDPSFLG ^b^	HLA-DRB1*07:01	0.7473454	20.49
ORF9 (N)	302	PQIAQFAPSASAFFG	HLA-DRB1*07:01	0.72358869	20.49
ORF2 (S)	24	LPPAYTNSFTRGVYY	HLA-DRB1*07:01	0.71760044	20.49
ORF2 (S)	306	FTVEKGIYQTSNFRV ^a,b^	HLA-DRB1*07:01 ^a^	0.70919003	20.49
ORF2 (S)	681	PRRARSVASQSIIAY ^b^	HLA-DRB1*07:01	0.67693256	20.49
ORF1ab	391	SGLKTILRKGGRTIA	HLA-DRB1*11:01	0.67477291	4.83

^a^ The binding between the peptide and HLA has been experimentally proven by T-cell assay and reported in IEDB; ^b^ The binding between the peptide and HLA has been experimentally proven by B-cell assay and reported in IEDB.

**Table 5 viruses-14-02328-t005:** List of epitopes that are fully conserved in SARS-CoV-2 ancestral sequence and four other Human CCCs. A total of two 9-mer epitopes originated from ORF1ab found to be 100% conserved in all five viruses.

Epitope Sequence	Immunogenicity	Identity (%)	HLA that Binds	Population Coverage (%)	Position	Location
HEFCSQHTM	0.03852	100	HLA-B*18:01, HLA-B*38:02, HLA-B*44:03	37.13	5202—5210	SARS-CoV-like RdRp [nsp12] Conserved polymerase motif E Putative RNA binding site
SLAIDAYPL	0.0619	100	HLA-A*02:01 ^a^	13.07	5254—5262	SARS-CoV-like RdRp [nsp12] Putative RNA binding site

^a^ The binding between the peptide and HLA has been experimentally proven by both T-cell assay and HLA ligand assay and reported in IEDB.

**Table 6 viruses-14-02328-t006:** Conservancy analysis of conserved epitopes in SARS-CoV-2 reference sequence against SARS-CoV-2 currently circulating VOCs of delta and omicron variants.

Start	Epitope Sequence	Alpha B.1.1.7 (158)	Beta B.1.351 (374)	Delta B.1.617.2 (1157)	Gamma P.1 (9)	Omicron BA.1.1 (884)
5202	HEFCSQHTM	100.00% (111/111)	100.00% (146/146)	100.00% (458/458)	100.00% (8/8)	100.00% (119/119)
5254	SLAIDAYPL	100.00% (111/111)	100.00% (146/146)	100.00% (458/458)	100.00% (8/8)	100.00% (119/119)

**Table 7 viruses-14-02328-t007:** Conservancy analysis of conserved Epitopes in SARS-CoV-2 against SARS-CoV and MERS-CoV which belong to the same Betacoronavirus family. The different residue is underlined.

SARS-CoV-2	SARS-CoV	MERS-CoV
^5202^HEFCSQHTM^5210^	^5179^HEFCSQHTM^5187^	^5188^HEFCSQHTL^5196^
^5254^SLAIDAYPL^5262^	^5230^SLAIDAYPL^5238^	^5239^SLAIDAYPL^5247^

**Table 8 viruses-14-02328-t008:** 9-mer epitopes originated from OR1ab with identical TCR contact residues between SARS-CoV-2 and Human CCC identified through IEDB Conservancy Analysis Tools, along with information on HLA that binds, population coverage, position and previous IEDB reports on the binding. The TCR contact residue of HLA Class I is located in position 3–8 (marked by underlined). The residue difference across the sequence is indicated with red color text.

Epitope Sequence ^[Domain Function]^	HLA that Binds ^[Population Coverage]^	229E ^[Position]^	NL63 ^[Position]^	HKU1 ^[Position]^	OC43 ^[Position]^
^5005^HL**MGWDYP**K^5013^ ^[RdRp (nsp12), Catalytic residue, Putative inhibitor binding site, Conserved polymerase binding A]^	HLA-A*11:01 ^a^ HLA-A*33:03 HLA-A*34:01 ^(62.81)^	**K**L**MGWDYP**K ^c^ ^(4676–4684)^	**M**L**MGWDYP**K ^b^ ^(4651–4659)^	**V**L**MGWDYP**K ^c^ ^(5066–5074)^	**V**L**MGWDYP**K ^c^ ^(4978–4986)^
^5251^FV**SLAIDA**Y^5259^ ^[RdRp (nsp12), Putative RNA binding site]^	HLA-A*34:01 HLA-B*15:02 HLA-B*15:21 HLA-B*35:05 ^(54.16)^	**Y**V**SLAIDA**Y ^b^ ^(4922–4930)^	**Y**V**SLAIDA**Y ^b^ ^(4897–4905)^	FV**SLAIDA**Y ^b^ ^(5312–5320)^	FV**SLAIDA**Y ^b^ ^(5224–5232)^
^6321^SI**VCRFDT**R^6329^ ^(Nsp14, N7-Mtase Active site)^	HLA-A*33:03 ^(29.13)^	SI**VCRFDT**R ^b^ ^(5981–5989)^	SI**VCRFDT**R ^b^ ^(5956–5964)^	SI**VCRFDT**R ^b^ ^(6378–6386)^	**AV****VCRFDT**R ^(6290–6298)^
^6844^KY**TQLCQY**L^6852^ ^(Coronavirus nsp13 super family)^	HLA-A*24:02 ^a^ HLA-A*24:07 ^(58.77)^	KY**TQLCQY****F ** ^b^ ^(6503–6511)^	KY**TQLCQY**L ^b^ ^(6474–6482)^	KY**TQLCQY**L ^b^ ^(6929–6937)^	KY**TQLCQY**L ^b^ ^(6842–6850)^

^a^ The binding between the peptide and HLA has been experimentally proven by HLA ligand assay and reported in IEDB. ^b^ Still binds to exactly the same allele with SARS-CoV-2 original epitope sequence according to netCTLpan with %Rank less than 1%. ^c^ Still binds to at least one same allele with SARS-CoV-2 original epitope sequence according to netCTLpan with %Rank less than 1%.

**Table 9 viruses-14-02328-t009:** 15-mer epitopes (9-mer core epitope is highlighted in bold) originated from ORF1ab with identical TCR contact residue between SARS-CoV-2 and Human CCC identified through IEDB Conservancy Analysis Tools, along with information on HLA that binds, population coverage, position and previous IEDB report on the binding. The TCR contact residue (highlighted in bold and underlined) of HLA Class II is located in position 2, 3, 5, 7 and 8 within the 9 mer core. The residue difference across the sequence is indicated with red color font.

Epitope Sequence ^[Position]^ ^[Domain Function]^	HLA ^[Population Coverage]^	229E ^[Position]^	NL63 ^[Position]^	HKU1 ^[Position]^	OC43 ^[Position]^
MNL**K****YAISAKNR**ART ^a^ ^[4934–4948]^ ^[RdRp (nsp12), Putative RNA binding site, Putative inhibitor binding site, Putative nsp7 interaction site, Conserved polymerase binding F]^	DRB1*11:01 ^[4.83]^	**L**NL**KYAISGKE****R**ART ^b^ ^[4605–4619]^	**L**NL**KYAISGKE****R**ART ^b^ ^[4580–4594]^	MNL**KYAISAKNR**ART ^b^ ^[4995–5009]^	MNL**KYAISAKNR**ART ^b^ ^[4907–4921]^
KAV**FISPYNSQN**AVA ^[5832–5846]^ ^[Nsp13, Helicase domain]^	DRB1*15:01 DRB1*15:02 DRB1*16:02 ^[51.25]^	KAV**FISPYNSQNY**VA ^c^ ^[5504–5518]^	KAV**FISPYNSQNY**VA ^c^ ^[5479–5493]^	**S**AV**FISPYNSQNY**VA ^c^ ^[5892–5906]^	KAV**FISPYNSQNFA**A ^c^ ^[5804–5818]^
AV**FISPYNSQNA**VAS ^[5833–5847]^ ^[Nsp13, Helicase domain]^	DRB1*15:01 DRB1*15:02 DRB1*16:02 ^[51.25]^	AVF**ISPY****N**S**QN****Y**VA**A**^c^ ^[5505–5519]^	AVF**ISPYNSQNY**VAS ^c^ ^[5480–5494]^	AVF**ISPYNSQNY**VA**K** ^c^ ^[5893–5907]^	AVF**ISPYNSQNFA**A**K** ^c^ ^[5805–5819]^
VFI**SPYNSQNAV**ASK^ a^ ^[5834–5848]^ ^[Nsp13, Helicase domain]^	DRB1*15:01 DRB1*15:02 DRB1*16:02 ^[51.25]^	VFI**SPYNSQN Y** **V**A**AR** ^[5506–5520]^	VFI**SPYNSQN Y** **V**AS**R** ^[5481–5495]^	VFI**SPYNSQN Y** **V**A**KR**^c^ ^[5894–5908]^	VFI**SPYNSQN F** **A**A**KR**^c^ ^[5806–5820]^
NCN**VDRYPANSI**VCR ^[6311–6325]^ ^[Nsp14, N7-Mtase active site]^	DRB1*15:01 DRB1*15:02 ^[47.35]^	NCN**VD M** **YPE F** **SI**VCR ^[5971–5985]^	NCN**VD M** **YPE F** **SI**VCR ^[5946–5960]^	NCN**VDK** **YPSNSI**VCR ^b^ ^[6368–6382]^	NCN**VDK** **YPPN A** **V**VCR ^[6280–6294]^

^a^ The binding between the peptide and HLA has been experimentally proven by B cell assay (ELISA) and reported in IEDB. ^b^ Still binds to exactly the same allele with SARS-CoV-2 original epitope sequence according to netMHCIIpan and exhibiting strong binding level with %Rank less than 1%. ^c^ Still binds to at least one same allele with SARS-CoV-2 original epitope sequence according to netMHCIIpan and exhibiting strong binding level with %Rank less than 1%.

**Table 10 viruses-14-02328-t010:** Cross-reactive CTL epitopes between SARS-CoV2 and BCG predicted by Blastp. The amino acid substitution is indicated by underlining and TCR contact residue (residue 3–8) is indicated by bold font. n.b. indicates that peptide was predicted as a non-binder to HLA allele.

SARS-CoV-2 Peptide	BCG Peptide	Protein Name in SARS-CoV-2	Protein Name in BCG	HLA Allele	Predicted HLA Binding Affinity ^a^	
					SARS-CoV-2	BCG
FV**FLVLLP**L	FV**FLALLP**F	ORF2	transmembrane ABC transporter ATP-binding protein	HLA-A*02:01	4.15	n.b
				HLA-A*34:01	10.54	11.56
				HLA-B*15:02	n.b	3.92
				HLA-B*15:13	n.b	4.93
				HLA-B*15:21	n.b	4.98
				HLA-B*18:01	n.b	9.84
				HLA-B*35:05	n.b	4.91
WL**IVGVAL**L	WL**MVGIAI**L	ORF3a	NUDIX domain-containing protein	HLA-A*02:01	3.97	3.53
				HLA-A*02:03	3.55	3.63
				HLA-B*38:02	n.b	13.42
	FL**IVGIEL**L	ORF3a	hypothetical protein N054_02474	HLA-A*02:01	3.97	2.96
				HLA-A*02:03	3.55	2.97
	WL**IAGVVF**L	ORF3a	membrane protein	HLA-A*02:01	3.97	2.78
				HLA-A*02:03	3.55	3.06
	WL**IVGYFA**V	ORF3a	MFS transporter	HLA-A*02:01	3.97	2.74
				HLA-A*02:03	3.55	2.99
SV**LLF****LAF**V	YVL**LFVSL**V	ORF4	cytochrome c biogenesis protein	HLA-A*02:01	3.34	3.72
KL**LEQWNL**V	KL**LEEWLA**M	ORF5	Isocitrate lyase	HLA-A*02:01	2.87	3.74
HL**VDFQVT**I	HL**VDFPVT**L	ORF6	RD1 region associated protein	HLA-A*02:01	4.40	4.41
	SL**VDSQVT**V	ORF6	transmembrane transport protein MmpL7	HLA-A*02:01	4.40	3.99
				HLA-A*02:03	n.b	3.52
FL**AFLLFL**V	FL**AFLDST**I	ORF7b	MFS transporter	HLA-A*02:01	2.34	3.68
				HLA-A*02:03	2.48	2.97
FY**SKWYIR**V	GY**EKWRIR**F	ORF8	SAM-dependent methyltransferase	HLA-A*24:02	6.18	7.60
				HLA-A*24:07	7.83	10.09
LL**LDRLNQ**L	YL**LDRLAE**L	ORF9	alpha-keto acid decarboxylase family protein	HLA-A*02:01	3.06	2.29
				HLA-A*02:03	3.50	2.45
	LL**LDRLPN**L	ORF9	cytochrome P450	HLA-A*02:01	3.06	2.81
				HLA-A*02:03	3.50	3.62
	LL**RDRLAQ**L	ORF9	inositol monophosphatase	HLA-A*02:01	3.06	n.b
				HLA-A*02:03	3.50	3.26

^a^ The unit of predicted HLA binding affinity score displayed is nM times 10^−5^.

**Table 11 viruses-14-02328-t011:** Cross-reactive HTL epitopes between SARS-CoV2 and BCG. The amino acid substitution is indicated by underlining. The core epitope is highlighted in red color and bold font indicating the TCR contact residue.

SARS-CoV-2	BCG	Gene Name in SARS-CoV-2	Protein Name in BCG	HLA Allele	Predicted HLA Binding Affinity *	
					SARS-CoV-2	BCG
VKHV Y**QL** R**A** R**SV**S PK	TDLL Y**QL** A**S** R**SV**S ST	ORF7a	amidase	HLA-DRB1*11:01	3.64	2.89
				HLA-DRB1*12:02	4.38	-

* The unit of predicted HLA binding affinity score displayed is nM times 10^−5^.

**Table 12 viruses-14-02328-t012:** Cross-reactive CTL epitopes between SARS-CoV2 and MMR vaccine. The amino acid substitution is indicated by underlining and TCR contact residue is indicated by bold font.

MMR	SARS-CoV-2 Peptides	MMR Peptides	Protein Name in SARS-CoV-2	Protein Name in MMR	HLA Allele	Predicted HLA Binding Affinity ^a^	
						SARS-CoV-2	MMR
Measles	SV**LLFLAF**V	YV**LLAVLF**V	ORF4	hemagglutinin	HLA-A*02:01	3.34	3.77
	IL**FLALIT**L	VM**FLSLIG**L	ORF7a	hemagglutinin	HLA-A*02:01	4.75	3.97
					HLA-A*02:03	-	4.31
	LL**LDRLNQ**L	RL**LDRLVR**L	ORF9	nucleocapsid protein	HLA-A*02:01	3.96	3.66
					HLA-A*02:03	3.50	-
Mumps	FV**F****LVLLP**L	FL**L****LILLP**L	ORF2	small hydrophobic protein	HLA-A*02:01	4.15	3.58
					HLA-A*34:01	10.54	-
	SV**LLFLAF**V	SLL**LFLLY**L	ORF4	small hydrophobic protein	HLA-A*02:01	3.34	4.67
	FL**AFLLFL**V	FL**TFLLLI**V	ORF7b	small hydrophobic protein	HLA-A*02:01	2.34	3.82
					HLA-A*02:03	2.48	3.73
Rubella	LL**LDRLNQ**L	AL**LDRLRG**V	ORF9	nonstructural capsid protein, partial	HLA-A*02:01	3.06	4.84
					HLA-A*02:03	3.50	4.84

^a^ The unit of predicted HLA binding affinity score displayed is nM times 10^−5^.

**Table 13 viruses-14-02328-t013:** Potential epitopes for universal vaccine construct. Listed are epitopes that are fully conserved across the coronavirus family and epitopes with identical TCR contact residue across SARS-CoV-2 and Human CCCs. Each of these epitopes was not similar to the human peptides.

Epitope Sequence	HLA that Binds	Immunogenicity	Population Coverage of Individual Epitope (%)	Similarity to Human Peptides
HEFCSQHTM	HLA-B*18:01, HLA-B*38:02, HLA-B*44:03	0.03852	37.13	None
SLAIDAYPL	HLA-A*02:01	0.0619	13.07	None
HLMGWDYPK	HLA-A*11:01, HLA-A*33:03, HLA-A*34:01	0.1053	62.81	None
FVSLAIDAY	HLA-A*34:01, HLA-B*15:02, HLA-B*15:21, HLA-B*35:05	0.00813	54.16	None
KYTQLCQYL	HLA-A*24:02, HLA-A*24:07	−0.06866	58.77	None
MNLKYAISAKNRART	DRB1*11:01	0.40189009	4.83	None
KAVFISPYNSQNAVA	DRB1*15:01, DRB1*15:02, DRB1*16:02	0.4720961	51.25	None
AVFISPYNSQNAVAS	DRB1*15:01, DRB1*15:02, DRB1*16:02	0.53704543	51.25	None
VFISPYNSQNAVASK	DRB1*15:01, DRB1*15:02, DRB1*16:02	0.22363359	51.25	None

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
