# Peer review of "Immunoinformatics Identification of the Conserved and Cross-Reactive T-Cell Epitopes of SARS-CoV-2 with Human Common Cold Coronaviruses, SARS-CoV, MERS-CoV and Live Attenuated Vaccines Presented by HLA Alleles of Indonesian Population"

_viruses, 2022, doi:10.3390/v14112328_

Round 1

Reviewer 1 Report

Gustiananda and colleagues have applied a bioinformatics approach to identify two fully conserved and nine heterologous SARS-CoV-2-derived T-cell epitopes with four human common cold coronaviruses (HCCC), using HLA alleles prevalent in the Indonesian population. The authors have performed several in silico experiments which could provide the foundation for further experimental studies. The topic is interesting and the manuscript is well-written and easy to read. However, several comments have to be addressed before the paper is accepted for publication.

 -        The title should be rephrased and become shorter. The names of the HCCC and the live attenuated vaccines can be mentioned in the Abstract and not the title.

-        Homology is not measurable, i.e., two sequences are either homologous or not. Therefore, xx% homology must change to xx% identity/similarity.

-        The date the sequences were retrieved should be provided in the manuscript.

-        The accession codes of the corresponding peptides should be also provided, for example, ORF1ab: YP_009724389.1?

-        The sequences of the Alpha (B.1.1.7), Beta (B.1.351), Delta (B.1.617.2), and Gamma (P.1) and Omicron (B. 1.1.529) SARS-CoV-2 variants could be provided as supplementary material.

-        The entire manuscript should undergo a thorough revision for syntax and grammar errors.

Line 69: "In order for T-cells to exhibit responses..." you can change to "elicit/prime/induce responses"

Line 92: "SARS-CoV-2 VOC were required and retrieved..." What do you mean by "required"?

Author Response

Responses to Reviewer 1 comments

We thank Reviewer 1 for the valuable comments, inputs and suggestions and we would like to address some of the comments and questions below.

Point 1

Reviewer 1: The title should be rephrased and become shorter. The names of the HCCC and the live attenuated vaccines can be mentioned in the Abstract and not the title.

M Gustiananda: Thank you for the suggestions.  Kindly find in the revised manuscript that we have rephrased the title to become shorter and deleted and moved the names of the HCCCs and live attenuated vaccines into the Abstract section.

Point 2

Reviewer 1: Homology is not measurable, i.e., two sequences are either homologous or not. Therefore, xx% homology must change to xx% identity/similarity.

M Gustiananda: Thank you for the correction. In the revised manuscript we have thoroughly changed the term ‘homology’ into ‘identity’ or ‘similarity’ when appropriate and paraphrased the sentence when necessary.

Point 3

Reviewer 1: The date the sequences were retrieved should be provided in the manuscript.

M Gustiananda: Thank you for the suggestion. In the revised manuscript we have provided the information regarding the date the sequences were retrieved.  We have also paraphrased the sentences as stated in the materials and methods section of the manuscript.

Point 4

Reviewer 1: The accession codes of the corresponding peptides should be also provided, for example, ORF1ab: YP_009724389.1?

M Gustiananda: Thank you for the suggestion, however, since the reference number such as NC_045512.2 has been provided, it will be redundant to provide the individual accession code for each protein.

Point 5

Reviewer 1: The sequences of the Alpha (B.1.1.7), Beta (B.1.351), Delta (B.1.617.2), and Gamma (P.1) and Omicron (B. 1.1.529) SARS-CoV-2 variants could be provided as supplementary material.

M Gustiananda: Thank you for the suggestion, we have provided the FASTA file of the sequences of the ORF1ab from Alpha (B.1.1.7), Beta (B.1.351), Delta (B.1.617.2), Gamma (P.1) and Omicron (B. 1.1.529) SARS-CoV-2 variants as supplementary material S1. 

Point 6

Reviewer 1: The entire manuscript should undergo a thorough revision for syntax and grammar errors.

M Gustiananda: Thank you for the suggestion, we have thoroughly revised the manuscript for syntax and grammar errors.

Point 7

Reviewer 1: Line 69: "In order for T-cells to exhibit responses..." you can change to "elicit/prime/induce responses"

M Gustiananda: Thank you for the correction.  We have changed accordingly

Point 8

Reviewer 1: Line 92: "SARS-CoV-2 VOC were required and retrieved..." What do you mean by "required"?

M Gustiananda: Thank you for the correction. It was a typo and has been revised accordingly.

Reviewer 2 Report

The paper has many abbreviations that are not defined when first used in the manuscript (HLA, BCG, OPV, MMR, so on). Abbreviation should be stated in complete form to help the readers’ ability to easily understand the manuscript.

Line 664: parenthesis should be omitted.

Line 683: Could the number of epitopes being the largest in ORF1ab be related to being the longest in the coronavirus genome? It seems that the spike gene has second largest number of epitopes and then follows the next largest gene. Could you elaborate why the number of epitopes matters when searching for the universal coronavirus vaccine target?

Author Response

Responses to Reviewer 2 comments

We thank Reviewer 2 for the valuable comments, inputs and suggestions and we would like to address some of the comments and questions below.

Point 1

Reviewer 2: The paper has many abbreviations that are not defined when first used in the manuscript (HLA, BCG, OPV, MMR, so on). Abbreviation should be stated in complete form to help the readers’ ability to easily understand the manuscript.

M Gustiananda: Thank you for the corrections. Kindly find in the revised manuscript that the abbreviations have been stated in complete form when first used.

Point 2

Reviewer 2: Line 664: parenthesis should be omitted.

M Gustiananda: Thank you for the correction. The parenthesis has been omitted as can be seen in line 665 of the revised manuscript.

Point 3

Reviewer 2: Line 683: Could the number of epitopes being the largest in ORF1ab be related to being the longest in the coronavirus genome? It seems that the spike gene has second largest number of epitopes and then follows the next largest gene. Could you elaborate why the number of epitopes matters when searching for the universal coronavirus vaccine target?

M Gustiananda: Thank you for the questions.  Yes, we agree with the reviewer comment that the number of epitopes will increase with the size of the proteins.  There has been a typo in the manuscript, since for the universal coronavirus vaccine target, it is not the number of epitopes per se that matters, but the number of ‘conserved’ epitopes is the most important one. Kindly find in the line 681 of the revised manuscript that the sentence has been paraphrased to include the word ‘conserved’.